# A Modular, Cost-Effective, and Pumpless Perfusion Assembly for the Long-Term Culture of Engineered Microvessels

**DOI:** 10.3390/mi16030351

**Published:** 2025-03-19

**Authors:** Shashwat S. Agarwal, Jacob C. Holter, Travis H. Jones, Brendan T. Fuller, Joseph W. Tinapple, Joseph M. Barlage, Jonathan W. Song

**Affiliations:** 1Department of Mechanical and Aerospace Engineering, The Ohio State University, Columbus, OH 43210, USA; agarwal.399@osu.edu (S.S.A.); jones.3318@osu.edu (T.H.J.); 2Department of Biomedical Engineering, The Ohio State University, Columbus, OH 43210, USA; holter.8@osu.edu (J.C.H.); fuller.441@buckeyemail.osu.edu (B.T.F.); tinapple.10@osu.edu (J.W.T.); 3The Comprehensive Cancer Center, The Ohio State University, Columbus, OH 43210, USA; 4Department of Biomedical Education and Anatomy, The Ohio State University, Columbus, OH 43210, USA; barlage.44@osu.edu

**Keywords:** modular microfluidics, xurography, hydraulic resistance, perfusion

## Abstract

Continuous perfusion is necessary to sustain microphysiological systems and other microfluidic cell cultures. However, most of the established microfluidic perfusion systems, such as syringe pumps, peristaltic pumps, and rocker plates, have several operational challenges and may be cost-prohibitive, especially for laboratories with no microsystems engineering expertise. Here, we address the need for a cost-efficient, easy-to-implement, and reliable microfluidic perfusion system. Our solution is a modular pumpless perfusion assembly (PPA), which is constructed from commercially available, interchangeable, and aseptically packaged syringes and syringe filters. The total cost for the components of each assembled PPA is USD 1–2. The PPA retains the simplicity of gravity-based pumpless flow systems but incorporates high resistance filters that enable slow and sustained flow for extended periods of time (hours to days). The perfusion characteristics of the PPA were determined by theoretical calculations of the total hydraulic resistance of the assembly and experimental characterization of specific filter resistances. We demonstrated that the PPA enabled reliable long-term culture of engineered endothelialized 3-D microvessels for several weeks. Taken together, our novel PPA solution is simply constructed from extremely low-cost and commercially available laboratory supplies and facilitates robust cell culture and compatibility with current microfluidic setups.

## 1. Introduction

Perfusion is integral to several microfluidic-based applications, such as to microphysiological systems (MPS) [1,2,3], single cell capture [4], mimicking perspiration [5,6], and on-chip sorting, positioning, and stimulation of *C. elegans* [7]. In the context of MPS and 3-D microtissues, perfusion ensures sufficient nutrient delivery, waste removal, and mechanical stimuli for sustaining long-term cell culture conditions for biological mechanistic and therapeutic screening studies. To meet these needs, researchers have commonly turned to externally controlled machines such as syringe pumps [8], peristaltic pumps [9,10], and rocker plates [11,12]. While these external controls have been widely implemented by microfluidic researchers, they are also subject to several operational challenges. For instance, syringe pumps and peristaltic pumps interface with microfluidic devices through tubing, which—depending on the material—can be prone to non-specific adsorption of analytes from the fluid being perfused and are subject to dead volume [13,14]. Moreover, insertion of tubing in microfluidic devices can introduce air bubbles, which often cause device failure by blocking channels and/or creating flow instabilities. Rocker systems provide a pumpless and tubeless approach for perfusion where flow occurs in microdevices resting on moving plates. However, the nature of the flow controlled by rocker plates is typically oscillatory and bidirectional, which may not be appropriate based on the physiology of interest, e.g., in blood microcirculation, where the intravascular flow under normal conditions is unidirectional and steady [15].

The challenges associated with equipment-based perfusion apparatuses have prompted researchers to develop easy-to-use perfusion setups that do not require an external power source and are tubeless [16,17,18,19,20,21]. For instance, a simple and commonly used approach is to insert a fluid-filled pipette tip [22] directly into the inlet port to create a hydrostatic pressure differential. However, in this setup, changes in pressure are transient and rapidly depleted, resulting in a temporally non-uniform flow that lasts for no longer than several seconds or minutes. Beebe and colleagues [23,24,25] proposed a passive pumping mechanism to create surface-tension-driven flow through the introduction of different sized droplets on the inlet and outlet ports of a microfluidic device. While this pumping platform is easy to implement, readily multiplexed, and compatible with conventional fluid handling, the pressure differential quickly equilibrates; it is also prone to flow control problems [26]. Komeya et al. [17] reported an interesting and bespoke pumpless design that uses a resistance microchannel between the fluid reservoir and the cell culture region of interest to sustain hydrostatic pressure-driven flow for up to several days. The resistance channel can be modified to control flow rates, and this design was used for long-term cultivation of endothelialized lumens [27]. However, this method requires fabrication of custom resistance microchannels using specialized facilities, such as a cleanroom, which may limit accessibility and translation to laboratories with no prior microfluidic and/or lithographic expertise.

The purpose of the present study is to develop a perfusion system that can benefit all biomedical researchers, including ones from non-microfluidic labs, interested in long-term cultivation of MPS. To this end, our solution should be easy to fabricate and implement, low-cost, accessible, and reliable in sustaining 3-D microtissues. Here, we present a pumpless perfusion assembly (PPA) that meets all these requirements. The PPA is assembled from components (or modules) consisting of sterile, pre-packaged (or “off-the-shelf”) syringes and syringe filters. These components are readily available from commercial suppliers and can be obtained at a total cost of USD 1–2 per PPA. The PPA retains the simplicity of gravity-based pumpless flow systems but incorporates high resistance filters that enable slow and sustained flow for long periods (from hours to days). The PPA is highly modular and can readily incorporate syringe filters of different properties reported by the commercial vendor (e.g., diameter, porosity, and pore size). We developed and validated our PPA approach by using several flow characterization schemes, where we observed that the flow rate of each PPA was a function of the syringe filter diameter and the pore size. We demonstrated that perfusion with the PPA can culture and sustain endothelialized 3-D microvessels for up to 21 days. Notably, we engineered 3-D microvessels employing an inexpensive and versatile rapid prototyping technique recently developed by our group using desktop craft cutting, or xurography [28]. Thus, the PPA provides a very simple, extremely low cost, and robust perfusion solution using commercial off-the-shelf components for the reliable culture of vascularized MPS, using laboratory methods that should be accessible to most labs worldwide.

## 2. Materials and Methods

### 2.1. Modular Pumpless Perfusion Assembly Using Ready-Made and Widely Available Laboratory Components

We constructed the two-component and modular pumpless perfusion assembly using standardized and aseptically packaged (i.e., sterile) laboratory consumables (Figure 1). The first component of the PPA is a trimmed syringe with the plunger removed, henceforth referred to as the liquid hopper. We used syringes of varying volume capacity (e.g., 20 mL and 60 mL) and trimmed to a height suitable for benchtop experimentation, incubation, or microscopy. The open-top nature of the liquid hopper enables efficient media exchange via pipetting. The second component is a hydrophilic syringe filter, which provides the main hydraulic resistive element of the PPA. Syringes were obtained from Henke Sass Wolf (Tuttlingen, Germany) and BD, (Becton Dickinson, Franklin Lakes, NJ, USA), and syringe filters containing semi-porous membranes of cellulose acetate (CA) were sourced from Tisch Scientific (North Bend, OH, USA). These CA syringe filters were selected because of their anti-fouling (i.e., low protein binding affinity) and hydrophilic properties consistent with typical cell culture media filtration [29].

The syringe filter of the PPA was pre-wetted by pushing 5 mL of an aqueous solution through the filter using an attached syringe and plunger (Figure 1A(i)). Sufficient pre-wetting is critical to eliminate any residual air trapped within the filter membrane—which would cause additional resistance—thus ensuring a continuous fluid circuit. For cell culture applications, pre-wetting can be achieved by cell-specific culture media or a buffered solution (e.g., PBS). Once the filter is properly wetted, the syringe is disconnected, and the filter is visually inspected for any air bubbles in the inlet or outlet of the syringe filter (Figure 1A(ii)). If present, air bubbles are pipetted out. The small size, pipette accessibility, and transparent polypropylene housing of the syringe filter aid in the visual inspection and removal of any air-bubbles. Alternatively, vacuum degassing can be used to evacuate air bubbles from the syringe filter. The PPA is fully assembled by connecting the liquid hopper to the pre-wetted syringe filter (Figure 1A(iii)). As standardized laboratory consumables, the syringe and syringe filter have Luer lock fittings, which ensures a leak-proof connection. The Luer slip outlet of the filter allows for insertion and press-fitting into a 4 mm diameter inlet port of a PDMS microfluidic device. Finally, liquid is added in the liquid hopper up to the desired pressure head height.

### 2.2. Resistance Characterization of the PPA Using Microfluidic Perfusion Experiments

We performed microfluidic experiments and analysis to determine the flow rates and resistances of the modular PPA. These experiments point to the objective of comparing our measured hydraulic resistances to the resistances which were calculated from the manufacturer’s specifications, henceforth referred to as manufacturer’s hydraulic resistances. We incorporated four different syringe filters in the PPA to characterize the modularity of this component: 0.22 μm and 0.45 μm pore sizes for both 4 mm and 13 mm diameter filters, henceforth denoted as 4_0.22_, 4_0.45_, 13_0.22_, and 13_0.45_. These specific filters were selected in part because we were able to obtain the corresponding manufacturer-specified pressure head (∆P) and flow rates (Q), which were used to calculate the manufacturer’s hydraulic resistances (R) (Table 1).

To experimentally obtain hydraulic resistances, henceforth referred to as measured hydraulic resistances, we measured the total pass-through volume through the microchannel of a PDMS microdevice delivered by the PPA over a discrete timeframe (Figure 1B). The single, straight-channel microfluidic device used in these experiments has been previously described by our group and was fabricated with PDMS soft lithography [30]. The height, width, and length of the microchannel in the silicon master mold is 150 µm, 500 µm, and 5 mm, respectively. The microchannel was replica molded by pouring PDMS pre-polymer onto the silicon wafers at a 10:1 base-to-curing-agent ratio and cured overnight in a 65 °C oven. Biopsy punches with 4 mm diameters were used to core the inlet and outlet ports of the microchannel. The PPA was attached to the inlet port of the straight channel microdevice pre-filled with double distilled water (ddH_2_O). A 20 mL syringe (BD, Franklin Lakes, NJ, USA, 302830) was chosen as the aspect ratio of the liquid hopper allowed for a significant hydrostatic pressure head (~100 mmH_2_O) to drive perfusion into the microchannel, in combination with a perceptible change in liquid meniscus height within the hopper. Moreover, a slit was made at the outlet port, and the device was submerged in a pool of ddH_2_O to establish a continuous fluid circuit and to minimize back pressure that would cause inaccurate measurements. The pass-through volume through the microchannel was determined from the drop in height of the ddH_2_O within the liquid hopper, photographs of which were taken at the beginning and end of experiments. The experimental time for each filter was informed by the results from our numerical calculations performed using the experimentally applied pressure head (100 mmH_2_O) and radius of the liquid hopper (9.125 mm, constructed from a 20 mL syringe) (Section 2.3) to achieve at least 300 µL of pass-through volume for each filter. Finally, the measured hydraulic resistance of each filter was determined using a closed-form solution of filter resistance as a function of volume drop and experimental time, described in Section 2.3.

### 2.3. Computational Prediction of Hydraulic Resistance and Volumetric Flow Rate

Volumetric flow rate can be written as proportional to the applied pressure and inversely proportional to the resistance of the system. In our setup, the applied pressure is due to the pressure head in the liquid hopper. The flow rate can then be written as follows:(1)$$Q=\frac{∆P}{R_{system}}=\frac{\rho g∆h}{R_{system}}$$
where *Q* is the volumetric flow rate and Δ*P* is the pressure differential between the inlet and outlet defined by ρ, the fluid density, *g*, the acceleration due to gravity, and ∆h, the height difference in the liquid meniscus between the inlet and outlet. The total hydraulic resistance of the system, $R_{system}$, is calculated by summing all resistances in series using a corresponding electrical circuit analogy as shown in Figure 1C:(2)$$R_{system}=R_{luer\;lock}+R_{filter}+R_{luer\;slip}+R_{microchannel}$$

$R_{luer\;lock}$ and $R_{luer\;slip}$ are calculated assuming Hagen–Poiseuille flow in a circular cross-section (e.g., a pipe) [31], which is obtained using the following equation:(3)$$R_{pipe}=\frac{128\mu L}{{\pi d}^{4}}$$
where *µ* is the dynamic viscosity of the fluid, *L* is the length of the pipe, and *d* is the inner diameter of the pipe. Based on Equation (3), $R_{luer\;lock}$ of our setup was 2.21 × 10^7^ Pa·s·m^−3^. Moreover, $R_{luer\;slip}$ for 4 mm and 13 mm filters were 3.94 × 10^7^ Pa·s·m^−3^ and 6.85 × 10^6^ Pa·s·m^−3^, respectively. Furthermore, we determined the resistance of our cylindrical microvessels to be 5.57 × 10^10^ Pa·s·m^−3^ from Equation (3).

The microchannel resistance, $R_{microchannel}$, was derived using analytical solutions corresponding to the microchannel geometry. For the simple rectangular microchannel we used the following equation [31]:(4)$$R_{microchannel}=\frac{12\mu L}{1-0.63\left(h/w\right)}\cdot \frac{1}{h^{3}w}$$
where *h* is the length of the short wall and *w* is the length of the long wall. For our straight-channel device, the microchannel resistance was estimated to be 3.90 × 10^10^ Pa·s·m^−3^. Moreover, since ${R_{microchannel}\gg \;R}_{luer\;lock}$ and ${R_{microchannel}\gg \;R}_{luer\;slip}$, Equation (2) can be reduced to:(5)$$R_{system}=R_{filter}+R_{microchannel}$$

Equation (5) can be substituted into Equation (1) to express flow rate, Q, in terms of the pressure head and the dominant hydraulic resistances of the system:(6)$$Q=\frac{\rho g∆h}{R_{filter}+R_{microchannel}}$$

Since the pressure head in the liquid hopper decreases with time, any flow rate measurement would be a time-averaged value. Hence, Equation (6) cannot be directly employed to compute measured hydraulic resistance for all filters tested. Instead, we sought to define the measured hydraulic resistance as a function of discrete pressure head measurements. This required calculating the instantaneous pressure head over time, where cumulative volume drop can then be quantified from the difference in heights given the cross-sectional area of the liquid hopper.

The rate of change in pressure head can be written as an ordinary differential equation:(7)$$\frac{dh}{dt}=-\frac{Q(t)}{A}$$
where *h* is the pressure head, *t* is time, *Q*(*t*) is the instantaneous flow rate, and *A* is the cross-sectional area of the liquid hopper. Substituting *Q* from Equation (6) into Equation (7) and recognizing that Equation (7) is of the form of an exponential decay equation, one can write the closed form solution as:(8)$$h(t)=h_{0}e^{-\frac{\rho gt}{A\left(R_{filter}+R_{microchannel}\right)}}$$
where *h*_0_ is the initial pressure head at time *t* = 0. A custom MATLAB (version 23.2.0.2515942 (R2023b) Update 7) script was written to compute the time dependence of fluid height within the liquid hopper and, by extension, the time-dependent flow rates for all syringe filters, using the manufacturer’s hydraulic resistances. These calculations were conducted to inform the choice of the syringe filter used for the long-term culturing of 3-D engineered microvessels.

To extract the measured filter hydraulic resistances from the experimentally observed pass-through liquid volumes, *R_filter_* was solved using Equation (8), leading to the following equation:(9)$$R_{filter}\left(∆V,∆t\right)=-\left[\frac{\rho g∆t}{A}\log^{-1}\left(1-\frac{∆V}{Ah_{0}}\right)\right]-R_{microchannel}$$
where Δ*V* is the experimentally observed pass-through liquid volume and Δ*t* is the experimental time duration.

Similarly, the time taken to reach a specific pressure head can be found using Equation (8). We chose to look at the time required for the liquid hopper to be depleted, which results in the following equation:(10)$$t_{empty}(A,R_{filter})=-\frac{A\left(R_{filter}+R_{microchannel}\right)}{\rho g}\log\left(\frac{h_{b}}{h_{0}}\right)$$
where *h_b_* is the height of the base of the liquid hopper. To generate a contour plot of time to deplete the fluid within the liquid hopper, an array of varying hopper radii ranging from 5 to 25 mm were used, along with an array of theoretical filter resistances ranging from 1 × 10^10^–1 × 10^13^ Pa·s·m^−3^. The bottom of the liquid hopper, *h_b_*, was measured to be 30 mm above the microdevice outlet.

### 2.4. Human Umbilical Vein Endothelial Cell Culture and Preparation

Commercially available human umbilical vein endothelial cells (HUVECs, Lonza, Basel, Switzerland) were purchased and maintained using endothelial cell growth medium-2 (EGM-2, Lonza). Cell passage numbers of 5–10 were used in this study. Cells were cultured in a humidified incubator at 37 °C and 5% CO_2_ with media exchange every two days. Cells were harvested by washing with 1x Dulbecco’s phosphate-buffered saline (PBS) without Mg/Ca (1x DPBS, Gibco, Waltham, MA, USA) followed by detachment using 0.05% Trypsin-EDTA 1x (Gibco, Waltham, MA, USA) for 3–4 min. Subsequently, trypsin was neutralized with 10% FBS in DMEM and cells were centrifuged at 200 RCF for 5 min. HUVECs were resuspended in media at a concentration of 50 × 10^3^ cells per μL in preparation to seed the lumens of the microfluidic devices.

### 2.5. Preparation of 3-D Engineered Microvessels and Integration of PPA for Long-Term Culture

We demonstrated the applicability of the PPA by enabling long-term culture of engineered microvessels constituted with HUVECs. To achieve this, we employed a xurography-based microfluidic device made from PDMS to engineer 3-D, cylindrical microvessels, as previously described by our group [28]. The device was constructed from 5 layers of PDMS bonded together by plasma oxidation and then bonded to a glass slide. A central channel was cut via xurography, and ports were formed by boring circular holes with biopsy punches. To ensure an adequate connection with the PPA, the top-most layer of the PDMS device was 6 mm in thickness with a 4 mm-diameter inlet port; this facilitated a press-fit connection between the PPA and the device. The schematic and photograph of the PPA with the microvessel-on-a-chip is shown in Appendix A.

After fabrication, microdevices were pre-coated with poly(dopamine) (PDA) to enhance collagen gel attachment to PDMS [32]. Type I collagen was added to the central ECM chamber at a concentration of 3 mg mL^−1^ (Corning, NY, USA, 354249) with a 250 µm-diameter nitinol wire running through the gel chamber from inlet to outlet port. Following collagen polymerization, the wire was removed, thereby leaving a hollow cylindrical lumen encased in hydrogel. Subsequently, 20 μL of fibronectin (Fisher Scientific, Waltham, MA, USA, FC010) was added to the lumen and incubated for at least 1 h at 37 °C. A total of 2 μL of HUVECs suspended in cell media at 50,000 cells/μL were seeded in the fibronectin-lined lumen. The devices were then placed in the cell incubator upside down for 30 min, followed by 15 min on each lateral side, before finally placing them in the normal upright orientation for 20 additional minutes. Subsequently, any unattached cells were washed with fresh media. Inlet and outlet ports were filled with cell media and the Petri dish was placed back into the incubator. The fibronectin coating coupled with the rotation steps ensured HUVECs attachment to the lumen surface. This was confirmed through phase contrast images of the top, midplane, and bottom planes of the microvessel, as shown in Appendix A. After 24 h of seeding the lumen with HUVECs, the PPA was filled with 5 mL of cell media and press-fitted into the inlet port of the microfluidic device to enable perfusion.

A slit was cut at the outlet port of the microdevice using a sterile cutting blade to achieve zero-gauge pressure at the outlet port. This is analogous to grounding in electrical circuits. A total of 100 μL of cell media was added to the slit to form a continuous fluid circuit. This enabled the pooling of perfused media from the slit onto the glass slide and surrounding Petri dish, thereby ensuring continuous flow and negligible backpressure. The PPA and the microdevice were housed in a deep Petri dish (Fisher Scientific, Waltham, MA, USA, FB0875711), and wrapped with parafilm (Fisher Scientific, Waltham, MA, USA, S37440) before placing in the 37 °C incubator, thereby minimizing contamination. The Petri dish surface was large enough to allow for pooling of the 5 mL of perfused media without any increase in fluid height. Perfused media was collected daily to prevent filling of the Petri dish. A humid environment was maintained with minimal evaporation due to the quantity of cell media present in the wrapped Petri dish. The presence of a humid environment was supported by the lack of any observed air bubble formation within the devices over the duration of the vessel culture. The syringe filter in the PPA was replaced daily to ensure precise flow rates. However, we suspect this was not necessary, and further testing would be needed to determine the maximum number of days a particular filter can maintain its flow rate profile. Phase contrast images of the microvessel were taken daily (Invitrogen, Waltham, MA, USA, EVOS XL Core). It is important to press fit the PPA in the microdevice with minimal force to prevent mechanical perturbation of the gel or microvessel.

### 2.6. Fluorescence Microscopy

We used a protocol developed by our group to fix and stain 3-D engineered microvessels [28]. Briefly, between day 7 and 14, cells in the microvessels were fixed with 4% paraformaldehyde (PFA) for 30 min at room temperature (RT), permeabilized with 0.2% Triton X-100 for 30 min at RT, blocked with blocking buffer for 30 min at RT, and subsequently stained for nuclei (DAPI, Sigma-Aldrich, St. Louis, MO, USA, D9542), VE-cadherin (monoclonal antibody, BD Biosciences, Franklin Lakes, NJ, USA, 561567), and F-actin (phalloidin, Fisher Scientific, Waltham, MA, USA, A12379). Microvessels were washed three times between each step with washing buffer 0.1% Tween 20 in 1x PBS. Blocking buffer consisted of 1% BSA in washing buffer. VE-cadherin was stained overnight at 4 °C, phalloidin was stained for 60 min at RT, and lastly DAPI was counterstained for 10 min. Images were taken using a Nikon A1R Live Cell Imaging Confocal Microscope controlled with NIS-Elements Software (version 6.02.03).

## 3. Results

### 3.1. Validation of the Hydraulic Resistances of Syringe Filters: Implications for Flow Modulation

First, we determined the range of hydraulic resistances for syringe filters integrated in the PPA, with the objective of comparing measured and manufacturer’s hydraulic resistances. Experiments were conducted with PPAs connected to microdevices containing a simple, straight microchannel (Figure 1B). We assessed filters of different diameters and pore sizes, which came with manufacturer-specified flow rates at a constant pressure head of 10 psi (Table 1). The flow rates and the pressure head were used to calculate the manufacturer’s hydraulic resistances of the syringe filters. We chose the following filters to study the effects of filter area and pore size on volumetric flow: 4_0.22_, 4_0.45_, 13_0.22_, and 13_0.45_. In addition, these filter pore sizes are widely available for purchase as they are commonly used for membrane filtration of particulates. For instance, 0.22 μm pore size membranes are the industry standard for sterile filtration of cell culture media to eliminate contaminating bacteria [33].

Using the manufacturer’s average hydraulic resistance values for each filter, we quantified the time dependence of the flow rate (Figure 2A) using the closed-form solution for instantaneous pressure head (Equations (6) and (8)). These calculations were performed using the initial experimentally applied pressure head (100 mmH_2_O) and radius of the liquid hopper (9.125 mm, constructed from a 20 mL syringe). The results informed the experimental time for measuring at least 300 µL of pass-through volume in each filter. Subsequently, we measured volume drop in the liquid hopper over the predetermined experimental time with the same initial pressure head for all aforementioned filters. The volume drop in the liquid hopper is equivalent to the pass-through volume, neglecting evaporation effects. Subsequently, Equation (9) was used to calculate the measured hydraulic resistances for all filters tested.

A comparative assessment of the measured and manufacturer’s hydraulic resistances is shown in Figure 2B. It can be observed that the measured hydraulic resistances mostly lie within the range of the manufacturers’ hydraulic resistance values. In addition, the ratio of measured resistances for 13 mm to 4 mm filters was found to be 13_0.22_:4_0.22_ = 0.102 for 0.22 µm pore size and 13_0.45_:4_0.45_ = 0.120 for 0.45 µm pore size. These ratio values closely match the inverse ratio of the manufacturer-listed filtration areas, A_4_:A_13_ = 0.115 (A_4_ = 0.125 cm^2^, A_13_ = 1.09 cm^2^). Furthermore, the ratios of measured filter resistances for 0.45 µm to 0.22 µm pore size for the same filter diameter were found to be 4_0.45_:4_0.22_ = 0.221 and 13_0.45_:13_0.22_ = 0.262. These ratio values closely match the inverse ratio of theoretical pore area A_0.22_:A_0.45_ = 0.239. Both observations are consistent with the notion that hydraulic resistance is inversely proportional to the cross-sectional area. Notably, we confirmed the resistance of these filters at much lower pressures (~10-fold lower) compared to the 10-psi pressure used by the manufacturer, indicating a linear dependence of the flow rate to the pressure head, as shown in Equation (1). These observations validated our approach to quantify hydraulic resistance as well as demonstrated that commercial syringe filters exhibit proportional resistances based on filtration area and pore size.

We also observed from the numerical calculations (Figure 2A) that for all filters there was a non-linear decay of flow rate with time. This non-steady flow rate is a consequence of a depleting pressure head within the liquid hopper, the rate of which depends on the radius of the hopper and the filter hydraulic resistance. It is important to point out that once the hopper is depleted, the air-liquid interface at the inner radius of the Luer lock (1 mm) causes the fluid to experience capillary arrest whereby the capillary force counteracts the remaining hydrostatic pressure achieving equilibrium and causing the flow to stop. One advantage of this phenomenon of capillary arrest is that the liquid hopper can be refilled directly with media, without having to disconnect the PPA from the microdevice to reestablish a continuous fluid circuit. The time taken for the liquid hopper to deplete, as given in Equation (10), was calculated for a range of hydraulic resistances and hopper radii, in the context of the PPA connected to our straight channel microdevice (Figure 2C). The contour plot illustrates the modular nature of our PPA, enabling researchers to modulate flow rate and, consequently, the flow rate decay, by swapping the syringe hoppers or syringe filters in the PPA. This notion is supported by the estimated time needed for flow arrest in a 60 mL syringe hopper, compared to a 20 mL syringe hopper, for each syringe filter. This outcome is due to the larger radius of the 60 mL syringe (13 mm) compared to the 20 mL syringe (9.125 mm), which results in a slower rate of head loss for the same instantaneous volumetric flow rate. Thus, the flow rate can be considered comparatively “steadier” for a liquid hopper with a larger radius. Moreover, increasing the filter membrane diameter or pore size decreases the overall filter resistance, resulting in more rapid head loss. Syringe filters with larger diameters and pore sizes would be better suited for short-term experiments where a high flow rate is required for only a short duration. In contrast, high resistance syringe filters are well suited for longer-term studies which require quasi-steady flow rate conditions.

Among the syringe filters considered, the 4_0.22_ filter has the highest hydraulic resistance. Consequently, it takes the longest time for liquid hopper depletion compared to all other chosen filters. Since flow rate is directly proportional to the pressure head, this also implies that the 4_0.22_ filter offers the steadiest flow among the filters tested. For instance, the fluid within the liquid hopper depletes in approximately 12 h for the PPA constructed from a 4_0.22_ filter and a 60 mL syringe (Figure 2C). In contrast, the same volume drop takes approximately 30 min for the 13_0.45_ filter and a 60 mL syringe. This positions the 4_0.22_ filter as having the “most steady” and longest-lasting flow rate among the four filters. This analysis informed our choice of using a 60 mL syringe hopper with a 4_0.22_ filter as the modular components of the PPA used for the long-term culture of endothelialized microvessels.

### 3.2. PPA Facilitates Long Term Culture of Engineered Microvessels for Several Days

Next, we employed a PPA with a 4_0.22_ filter and 60 mL syringe for long-term culture of HUVEC-lined microvessels. The PPA was press-fitted into the inlet port of the microdevice 24 h after seeding (Figure 3A). Figure 3B shows the theoretical instantaneous height of cell culture media in the liquid hopper, using the manufacturer’s average hydraulic resistance for each tested filter. The height of media in the liquid hopper depletes to the minimum height fastest for 13_0.45_ and slowest for 4_0.22_ due to the relative hydraulic resistances. The minimum pressure head in Figure 3B corresponds to the elevation of the base of the liquid hopper, at which point capillary arrest would occur.

We also calculated the theoretical instantaneous flow rate and wall shear stress within the microengineered vessel (Figure 3C), using the manufacturer’s hydraulic resistances for the syringe filters. It was observed that 4_0.22_ has the lowest flow rate and is the only filter with shear stress values below 1 dyn/cm^2^. Although 3 dyn/cm^2^ is typically considered physiological for HUVECs, minimal shear stresses (~0.1 dyn/cm^2^) are sufficient to maintain HUVEC viability [34]. Ultimately, we selected the 4_0.22_ filter for integration into the PPA for two principal reasons. First, as mentioned above, it takes the longest time to deplete the liquid hopper and needs replenishment only once in ~12 h (neglecting evaporation effects in a sufficiently hydrated Petri dish placed inside a 37 °C cell culture incubator). Second, since the resistance of the 4_0.22_ filter is an order of magnitude higher than the cylindrical microvessel (comparable to typical resistances for MPS), it effectuates the “most steady” flow rate of the syringe filters tested, as shown in Figure 3C. The instantaneous flow rate results demonstrate only a 4.5% decrease in flow rate over the first 60 min for the 4_0.22_ filter. These aspects reinforced our decision to use the 4_0.22_ filters for the long-term culturing of HUVEC-lined microvessels.

Phase contrast images of a microvessel perfused using this PPA are shown in Figure 3D for Days 1, 5, and 11. The longest duration for which we cultured the HUVEC microvessels was 21 days. Following immunofluorescent staining for VE-cadherin, DAPI and actin, we confirmed with confocal microscopy that the perfused engineered microvessel was stable with barrier integrity and with an open circular lumen (Figure 3E,F). The average thickness of the endothelial monolayer was 7.1 μm, which was determined by analyzing the x–y cross-sectional profiles within the confocal z-stack corresponding to the midplane of the actin-stained microvessel. This agrees with the endothelial layer thickness reported in the literature [35]. Interestingly, actin fibers in HUVECs did not appear elongated or aligned in the flow direction. This result can be attributed to the sub-physiological levels of average wall shear stress values ($\tau_{w}$ < 1 dyn/cm^2^) for 4_0.22_ as shown in Figure 3C. It has been observed previously that HUVECs lack any actin stress fibers when exposed to 0.8 dyn/cm^2^ of shear stress [36].

## 4. Discussion

The limitations of equipment-based perfusion apparatuses prompted us to develop a novel perfusion solution for microfluidic culture applications that meet the following design considerations. First, the flow rates produced by the perfusion setup can be sufficiently well characterized, are predictable, and fall within desirable ranges over a prescribed timeframe for the intended applications. Second, the setup is low cost without the requirement for the procurement of specialized equipment. Third, the perfusion apparatus is portable, does not require external power or control, and has a minimal footprint such that the entire system (both the microdevice and perfusion apparatus) fits within a Petri dish and can be placed in the same controlled environment (e.g., cell culture incubator). Fourth, the setup is associated with lower operational challenges, enabling researchers to pursue perfusion studies more readily. Fifth, the setup is amenable to conventional fluid handling techniques (e.g., pipetting), which may foster scaling up for high-throughput applications and foster adoption by scientists with no prior microfluidic and microfabrication training. Our study puts forth the described PPA that meets all these design considerations.

Our results provide ample support for the PPA to meet the first design consideration. We have shown that the experimentally determined hydraulic resistances of the four tested filters fall within the range of the manufacturer’s specifications. This allows researchers to use the provided product specifications to reliably predict flow rate and duration of flow for their specific experimental setup. In addition, since the flow rate and the duration of flow are functions of both components of the PPA—specifically the filter hydraulic resistance and liquid hopper cross-sectional area—users have flexibility to select the filter and syringe that best meet their experimental design requirements. To demonstrate this point, we utilized a 4_0.22_ filter and a 60 mL syringe for long-term perfusion of an engineered microvessel, enabling culture and maintenance of HUVEC vessels for up to 21 days.

Regarding the second design consideration, part of the ambition of this study was to develop a cost-efficient and functional perfusion system that facilitates access to 3-D microfluidic perfusion studies. A recent example that meets these needs is a 3D-printed mini-peristaltic pump, which can be fabricated at a fraction of its commercial counterparts (~USD 175) and leverages open-source programmable electronic controllers [10]. Comparatively, we offer a perfusion solution that is incredibly low cost (USD 1–2 per PPA) and requires no additional capital investment. Indeed, the components used to construct the PPA (syringe and syringe filter) are often presently stocked in most wet labs. Hence, this status may enhance research workflows because an effective PPA can be constructed without waiting for the procurement of materials or equipment installation. Notably, our solution is devoid of any external machines or controllers. The versions of the PPA described in this study used only readily available and prepackaged (or “off-the-shelf”) products, thereby employing favorable economies of scale to suppress costs. Nonetheless, our PPA solution also allows for a degree of customization as several manufacturers of syringe filters may be willing to work with researchers to fabricate membranes of different materials, thicknesses, diameters, and pore sizes. All these parameters will have an impact on the effective hydraulic resistance of the syringe filter, which we showed to be the dominant resistive term that determines the flow rates in an integrated PPA-microfluidic setup, for all tested filters. Our study also describes a generalizable microfluidic-based characterization scheme using experiments and numerical calculations. This scheme can be used by others for deriving the perfusion characteristics of other PPAs, including ones that integrate custom syringe filter modules.

A leading cause of detrimental air bubble formation is changes in temperature and pressure between a microfluidic device residing inside an incubator and a perfusion controller placed outside of an incubator and connected via tubing. Perfusion controllers (e.g., a syringe pump) often cannot be placed inside an incubator for various reasons. One potential reason is the heat dissipation from these controllers when operating, which disrupts effective temperature control within an incubator. Another reason is that perfusion controllers are typically much larger than the microfluidic device and therefore occupy valuable shelf space within an incubator or simply may not fit within the incubator altogether. In contrast, our PPA is only slightly larger than a microfluidic device, thereby enabling us to readily meet the third design requirement. We note that the PPA does add some vertical height to the setup that may be of consideration during microscopy. Nonetheless, as a highly portable and tubeless solution, our PPA intrinsically circumvents the pain points that can cause air bubble formation and experiments to fail. For long-term culture experiments with 3-D engineered microvessels, we used a PPA constructed with the 4_0.22_ syringe filter and a 60 mL syringe hopper. The perfusion characteristics of this PPA required us to replenish the PPA only once every 24 h by adding 5 mL of culture media. Thus, our perfusion solution readily meets the fourth design requirement by posing little to no technical barriers for researchers conducting experiments that culture engineered microvessels. Finally, regarding the fifth design requirement, the open top of the PPA is amenable to fluid handling—from simple pipetting to scaled-up automation. This is advantageous because the liquid hopper can be continuously refilled to accommodate steady flow into the microvessel by maintaining a constant pressure head in the liquid hopper. Moreover, the PPA can be applied to microvessel systems to model intravenous infusion of drugs or inhibitors during therapeutic screening studies.

The described PPA provides a robust approach for enabling long-term cell culture and continuous supply of nutrients and waste removal in MPS. Yet, we recognize that there are several important limitations of the PPA as a microfluidic perfusion solution. First, like most gravity-based perfusion systems, the flow rates of the PPA vary temporally. For the 4_0.22_ syringe filter, we observed only a 4.5% decrease in flow rate over 60 min. However, certain applications may require narrower fluctuations in flow rates over time. Second, the range of time-averaged flow rates offered by the PPA may be too narrow for certain desired applications. Along these lines, the maximum shear stress levels of 0.65 dyn/cm^2^ that were provided by the PPA in the engineered microvessel studies were much lower than physiological levels (typically > 3 dyn/cm^2^) [34]. This notion was reinforced by the lack of characteristic endothelial cell alignment and elongation in response to physiological shear stress levels observed in our studies. Our group and others have deep interests in studying the effects of physiological shear stress on endothelial cell morphogenesis, angiogenesis, and vessel remodeling [37,38]. Hence, external pumps and rocker plates may remain the mainstay flow applicators for these types of microfluidic studies. Although we demonstrated the effectiveness of the PPA for long-term culture of HUVECs, the characteristics of the PPA may be applicable for cell culture applications where low shear stress conditions are desirable, such as what was shown previously with CaCo-2 cells [21]. Collectively, unlike most microfluidic perfusion systems, researchers will encounter minimal technical and operational barriers and only nominal monetary cost with the PPA. Hence, we believe these characteristics of the PPA may help enhance the accessibility of MPS, especially to those with no prior microfluidic or microfabrication training.

## 5. Conclusions

This paper presents a pumpless perfusion assembly obtained by the integration of low-cost, standardized and aseptically packaged laboratory consumables. We demonstrated that the PPA enabled predictable flow rates based on theoretical calculations of the total hydraulic resistance of the assembly and experimental characterization. Finally, we showed that the PPA enabled reliable long-term culture of HUVEC-lined engineered microvessels for several weeks. While these results show the capability of our PPA, we anticipate the fabrication of custom PPAs to expand their versatility for microfluidic applications. For instance, the dimensions of the hydraulic resistor can be modified to allow for the passage of cells, thereby enabling studies involving perfusion of cells (e.g., immune cells) suspended in media [39]. Moreover, the custom PPAs can also be optimized to minimize pressure surge [40] which is experienced by the microchannel while press-fitting the PPA in the inlet port of the device. Nevertheless, we posit that the value proposition of our PPA as a microfluidic perfusion solution is very high due to its ease of construction from extremely low-cost and commercially available laboratory supplies, accessibility, robust culture, and compatibility with current MPS setups.

## 6. Patents

A provisional US patent for this idea has been filed through the Ohio State University’s Technology and Commercialization Office.

## Figures and Tables

**Figure 1 micromachines-16-00351-f001:**
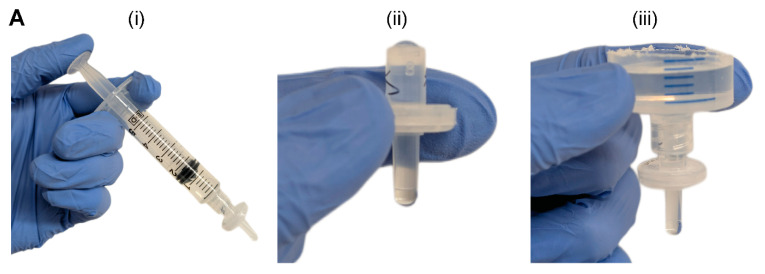
Pumpless perfusion assembly (PPA). (**A**) Detailed stepwise instructions for modular PPA: (**i**) the syringe filter is pre-wetted, (**ii**) the syringe filter is inspected for air bubbles, (**iii**) the liquid hopper is attached to wetted syringe filter to form the PPA. (**B**) Schematic of the PPA press-fitted into the inlet port of a straight-channel PDMS microdevice illustrating the gravity-based pressure head (P_1_–P_2_), which drives fluid flow. A slit was cut at the outlet to allow pooling of perfused liquid. Dashed region represents the top view of the straight microchannel. (**C**) Equivalent electrical circuit analog for the schematic shown in (**B**) demonstrating the hydraulic resistances in the system.

**Figure 2 micromachines-16-00351-f002:**
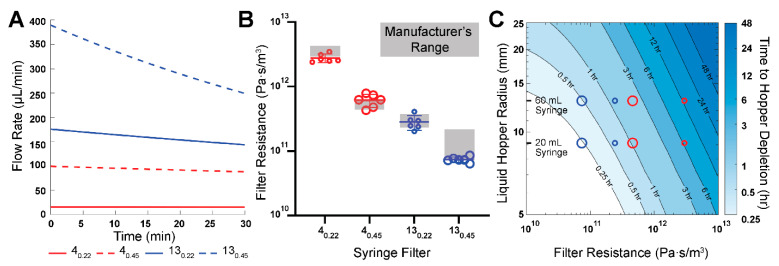
The measured and the manufacturer’s hydraulic resistances of syringe filters. (**A**) Instantaneous flow rate over the first 30 min for an initial 100 mmH_2_O pressure head and a 20 mL syringe as the liquid hopper. Flow rate values calculated with Equations (6) and (8) using the manufacturer’s hydraulic resistances. (**B**) Comparative assessment of the measured and manufacturer’s hydraulic resistances for the four types of syringe filters. (**C**) Contour plot visualizing time taken for depletion of fluid within the liquid hopper for a range of syringe filter resistances and hopper radii based on Equation (10) and a 40 mmH_2_O pressure head. Small and large, red and blue circles correspond to syringe filter labels from panel B.

**Figure 3 micromachines-16-00351-f003:**
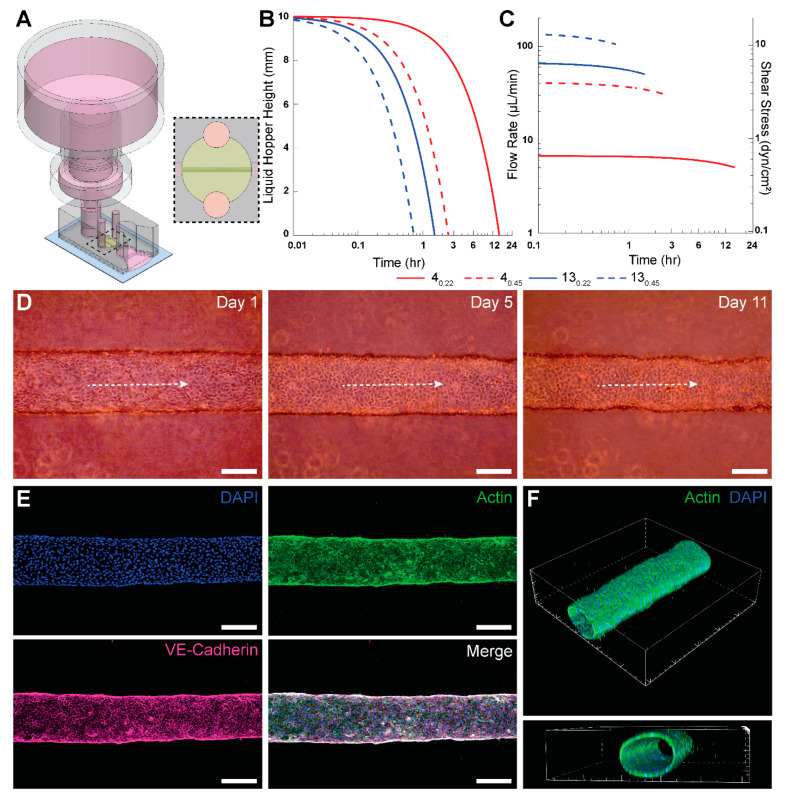
Implementation of the PPA for long-term culture of 3-D engineered microvessels. (**A**) Isometric view of PPA constructed from a trimmed 60 mL syringe and a 4_0.22_ filter press-fitted into a PDMS-based device with a cylindrical HUVEC-lined microvessel. A slit was cut at the outlet to allow pooling of perfused cell culture media. Dashed region represents the top view of the perfused microvessel encapsulated in collagen (yellow: collagen, pink: cell culture media, green: endothelial cell-lined lumen, blue: glass slide, dark gray: PDMS microdevice, and light gray: polypropylene housing of the PPA). (**B**) Theoretical instantaneous media height within the liquid hopper for the PPA consisting of modular syringe filters presented in Table 1 connected to the microvessel device as shown in A. (**C**) Theoretical instantaneous flow rates for the PPA and the directly proportional intravascular wall shear stress within a 250-μm-diameter cylindrical microvessel. (**D**) Phase contrast images of a representative microvessel at Day 1, 5 and 11. Dashed white arrows indicate intravascular flow direction. (**E**) Confocal z-projection of an intact cylindrical microvessel fully lined with HUVECs. The vessel was fixed at day 7, and stained for F-actin (phalloidin, green), VE-cadherin (monoclonal antibody, pink), and nuclei (DAPI, blue). (**F**) The 3-D renders of the microvessel from confocal z-stack to show cellular monolayer and patency. Scale bars in (**D**,**E**) are 200 μm. The interval of major ticks in (**F**) is 200 μm.

**Table 1 micromachines-16-00351-t001:** Manufacturer’s hydraulic resistances. Values for filter diameter, pore size, flow rate, and pressure head were obtained from the manufacturer’s specifications.

Filter Diameter (mm)	Pore Size (µm)	Flow Rate (Q) (@∆P = 10 psi) (mL/min)	Resistance (R) * (Pa·s/m^3^)
4	0.22	1.0–1.4	$3.54\pm 0.59\times 10^{12}$
4	0.45	7–9	$5.25\pm 0.66\times 10^{11}$
13	0.22	11–17	$3.09\pm 0.66\times 10^{11}$
13	0.45	19–56	$1.45\pm 0.72\times 10^{11}$

* Calculated using R = Δ*P*/Q, Equation (1).

## Data Availability

The data are available as per request.

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
