# Peer review of "A Modular, Cost-Effective, and Pumpless Perfusion Assembly for the Long-Term Culture of Engineered Microvessels"

_micromachines, 2025, doi:10.3390/mi16030351_

Round 1

Reviewer 1 Report

Comments and Suggestions for Authors

This paper proposes a pumpless perfusion assembly (PPA) that uses gravity drive and high-resistance filters to achieve long-term stable microfluidic perfusion. This method reduces the reliance on complex microfluidic equipments, such as syringe pumps and peristaltic pumps, and simplifies the experimental design of microvascular culture. The modular design of this method allows researchers to adjust the flow rate by changing the filter or syringe size, which provides flexibility for different biological research applications. The manuscript is recommended for publication in micromachines. Some comments are suggested below.
1. Quantitative analysis data next to Figure 3D, such as cell viability and layer thickness, are suggested.
2. Some papers on microfluidics are suggested, such as Mater. Today Nano, 16, 100136, (2021) and Lab chip, 16, 1346, (2016).
3. In the conclusion section, future optimization strategies should be discussed.
4. Figure 2 is out of the border.
5. There are some typos, such as two "." in line 89. What is the volume of reference 36?

Comments on the Quality of English Language

The english expression could be improved. For example, “Our solution is a modular pumpless perfusion assembly (PPA), which is constructed from commercially available, interchangeable, and aseptically packaged syringes and syringe filters.” could be changed to “We propose a simple yet effective modular pumpless perfusion assembly (PPA), assembled from commercially available sterile syringes and syringe filters, ensuring easy accessibility and cost-effectiveness.”.

Author Response

This paper proposes a pumpless perfusion assembly (PPA) that uses gravity drive and high-resistance filters to achieve long-term stable microfluidic perfusion. This method reduces the reliance on complex microfluidic equipments, such as syringe pumps and peristaltic pumps, and simplifies the experimental design of microvascular culture. The modular design of this method allows researchers to adjust the flow rate by changing the filter or syringe size, which provides flexibility for different biological research applications. The manuscript is recommended for publication in micromachines. Some comments are suggested below.

We are grateful to the reviewer for positive sentiment of our manuscript and for the valuable suggestions for improving the manuscript. Below is our composed response:

  1. Quantitative analysis data next to Figure 3D, such as cell viability and layer thickness, are suggested.
    We thank the reviewer for this recommendation. We have utilized the confocal images of the microvessels to determine the endothelial layer thickness. This result is reported on page 10, line 407:

    The average thickness of the endothelial monolayer was 7.1 μm, which was determined by analyzing the x-y cross-section profiles within the confocal z-stack corresponding to the midplane of the actin-stained microvessel. This agrees with the endothelial layer thickness reported in literature [35].

  2. Some papers on microfluidics are suggested, such as Mater. Today Nano, 16, 100136, (2021) and Lab chip, 16, 1346, (2016).
    We thank the reviewer for suggesting these papers. We reviewed both these and have added the Mater Today Nano citation in the revised manuscript. The second manuscript (Lab Chip) is on droplet-based microfluidics formed using multi-inlet/multi-phase microsystems. Since our application demonstrated in this manuscript is for a single inlet culture model, we elected to not cite the second manuscript.

  3. In the conclusion section, future optimization strategies should be discussed. Figure 2 is out of the border.
    We thank the reviewer for this recommendation. The conclusion section of the manuscript has been revised accordingly and is highlighted in yellow (Page 13, line 530). We have added the following sentences in the conclusion section to follow through on this suggestion:

    While these results show the capability of our PPA assembly, we anticipate the fabrication of custom PPAs to expand their versatility for microfluidic applications. For instance, the dimensions of the hydraulic resistor can be modified to allow for the passage of cells, thereby enabling studies involving perfusion of cells (e.g. immune cells) suspended in media [39]. Moreover, the custom PPAs can also be optimized to minimize pressure surge [40] which is experienced by the microchannel while press-fitting the PPA in the inlet port of the device.

    Regarding Figure 2, we have uploaded all Figures as part of the manuscript submission process and kindly request the Editor ensures Figures and Equations are printed within the document borders. For instance, the legend on the right side of Figure 2C is missing from the manuscript typeset document provided by the journal.

  4. There are some typos, such as two "." in line 89. What is the volume of reference 36?
    We thank the reviewer for catching these errors. These typos have been corrected in the revised manuscript. We have also added the volume of reference 36.

Reviewer 2 Report

Comments and Suggestions for Authors

The paper describes a very simple but useful approach for organizing continuous perfusion in microfluidic devices for long term incubation of cells. Due to its simplicity it might be very useful for the community. The paper is clearly written and has a good presentation. However I have two questions:

1) It is not clear from the images how is the outlet from the chip organized. It is mentioned in the paper that during long time incubations 5 ml of culture medium is added daily. How the reservoir for such a volume of waste medium is organised? Please show it on Image 3A or add it into the supplementary.

2) PDMS has high gas permeability, therefore, during long time incubations the medium can evaporate in the channels of a PDMS microfluidic device leading to air bubbles formation. Have you encountered such a problem? Do you somehow increase the humidity in the incubator?   

Author Response

The paper describes a very simple but useful approach for organizing continuous perfusion in microfluidic devices for long term incubation of cells. Due to its simplicity it might be very useful for the community. The paper is clearly written and has a good presentation. However I have two questions:

We appreciate the reviewer’s positive feedback and for the insightful questions. Below is our composed response.

  1. It is not clear from the images how is the outlet from the chip organized. It is mentioned in the paper that during long time incubations 5 ml of culture medium is added daily. How the reservoir for such a volume of waste medium is organised? Please show it on Image 3A or add it into the supplementary.
    We recognize the reviewer’s concern for the description of the outlet being less clear. We have revised the manuscript text to enhance its clarity. Specific to the reviewer’s concern, the text has been revised in section 2.5 (Page 7, line 263):

    A slit was cut at the outlet port of the microdevice using a sterile cutting blade to achieve zero-gauge pressure at the outlet port. This is analogous to grounding in electrical circuits. 100 μL of cell media was added to the slit to form a continuous fluid circuit. This enabled pooling of perfused media from the slit onto the glass slide and surrounding petri dish, thereby ensuring continuous flow and negligible backpressure. The PPA and the microdevice were housed in a deep petri dish (Fisher Scientific, FB0875711), and wrapped with parafilm (Fisher Scientific, S37440) before placing in the 37°C incubator, thereby minimizing contamination. The petri dish surface was large enough to allow pooling of the 5ml of perfused media without any increase in fluid height. Perfused media was collected daily to prevent filling of the petri dish.
  1. PDMS has high gas permeability, therefore, during long time incubations the medium can evaporate in the channels of a PDMS microfluidic device leading to air bubbles formation. Have you encountered such a problem? Do you somehow increase the humidity in the incubator?
    The reviewer is right to point out that during long time incubations the medium can evaporate in the channels of a PDMS microfluidic device leading to air bubbles formation. We thank the reviewer for asking this question. To follow up on this question, we have added the following text in section 2.5 (page 7, line 272):

    A humid environment was maintained with minimal evaporation due to the quantity of cell media present in the wrapped petri dish. The presence of a humid environment was supported by the lack of any observed air bubble formation within the devices over the duration of vessel culture.

Reviewer 3 Report

Comments and Suggestions for Authors

This manuscript introduces a simple and cost-effective pumpless perfusion assembly (PPA) for long-term microvessel culture. The idea is innovative and could be a valuable tool for labs that don’t have access to complex perfusion systems. The study is well-organized, and the experiments are clearly explained. However, a few areas need more clarification and detail to improve the manuscript’s clarity and impact.

  1. Page 3, Line 112: The authors mention inspecting air bubbles visually during pre-wetting, which can be unreliable. Can you suggest a more consistent method, like vacuum degassing or controlled flow to remove air bubbles? Since trapped air can interfere with perfusion, this is an important step to get right.
  2. Page 7, Line 237: The microchannel is a key part of this work, but its fabrication is only briefly mentioned. Since not all readers will be familiar with the referenced protocol, adding more details in the main text or supplementary material would be helpful.
  3. The manuscript does not explain how the hydrogel solution stays confined in the central chamber without leaking into the inlet or outlet. Given that narrow PDMS connections exist between these areas, capillary effects might draw liquid into unwanted spaces. Even if a rod or wire is used, sealing completely could be tricky. Could you provide more details on how leakage is prevented?
  4. The paper does not give enough information about how HUVECs are seeded and how their attachment is ensured. Since the microchannel is relatively large, it could be difficult for cells to attach to the top surface. More details on how this issue was addressed would be useful. Including images showing cell attachment and growth from Day 0 to Day 5 would strengthen this section.
  5. The inlet setup is well described, but what about the outlet? How is pressure controlled at the outlet? Is there a method to prevent backflow and keep the system running smoothly? Also, what steps were taken to prevent contamination over long culture periods?

Author Response

This manuscript introduces a simple and cost-effective pumpless perfusion assembly (PPA) for long-term microvessel culture. The idea is innovative and could be a valuable tool for labs that don’t have access to complex perfusion systems. The study is well-organized, and the experiments are clearly explained. However, a few areas need more clarification and detail to improve the manuscript’s clarity and impact.

We appreciate that the reviewer found our idea innovative and the experiments to be clearly explained. We are also grateful to the reviewer for their valuable suggestions for improving the manuscript. Below is our composed response.

  1. Page 3, Line 112: The authors mention inspecting air bubbles visually during pre-wetting, which can be unreliable. Can you suggest a more consistent method, like vacuum degassing or controlled flow to remove air bubbles? Since trapped air can interfere with perfusion, this is an important step to get right.
    We wholly agree with the reviewer that visual inspection of air bubbles is an important step to get right. Pre-wetting with 5 mL of liquid usually did not leave any air bubbles in most devices. We refer the reviewer to the revised text in section 2.1 (page 3, line 112), highlighted in yellow:

    The small size, pipette accessibility, and transparent polypropylene housing of the syringe filter aid in the visual inspection and removal of any air-bubbles. Alternatively, vacuum degassing can be used to evacuate air bubbles from the syringe filter.

  2. Page 7, Line 237: The microchannel is a key part of this work, but its fabrication is only briefly mentioned. Since not all readers will be familiar with the referenced protocol, adding more details in the main text or supplementary material would be helpful.
    We thank the reviewer for this suggestion. We have added Supplementary Figure 1 to visualize the PPA and the individual layers of the device. We have also added text to section 2.5 (page 7, line 244), highlighted in yellow:

    The PPA and the individual layers of the device are shown in Supplementary Figure 1.
  1. The manuscript does not explain how the hydrogel solution stays confined in the central chamber without leaking into the inlet or outlet. Given that narrow PDMS connections exist between these areas, capillary effects might draw liquid into unwanted spaces. Even if a rod or wire is used, sealing completely could be tricky. Could you provide more details on how leakage is prevented?
    We recognize the reviewer’s concern for the capillary effects drawing unpolymerized hydrogel into the inlet and outlet ports. In our microdevice, when the unpolymerized hydrogel is added into the gel port, it occupies 1) the central ECM chamber region, and 2) the region around the nitinol wire in the xurography cut microchannel connecting the inlet/outlet ports with the central ECM chamber. The filling of the hydrogel in the latter scenario is aided by capillary effect. Moreover, the hydrogel viscosity imparts the ability to fill the hydrogel only in regions 1 and 2. Thus, we do not observe any leakage into the inlet and outlet ports. 
  1. The paper does not give enough information about how HUVECs are seeded and how their attachment is ensured. Since the microchannel is relatively large, it could be difficult for cells to attach to the top surface. More details on how this issue was addressed would be useful. Including images showing cell attachment and growth from Day 0 to Day 5 would strengthen this section.
    We recognize the reviewer’s concern for attachment of HUVECs. We have added the following text in section 2.5 (page 7, line 251), highlighted in yellow:

    Subsequently, 20 μL of fibronectin (Fisher Scientific, FC010) was added to the lumen and incubated for at least 1 hour at 37°C. 2 μL of HUVECs suspended in cell media at 50,000 cells / μL were seeded in the fibronectin-lined lumen. The devices were then placed in the cell incubator upside down for 30 minutes followed by 15 minutes on each lateral side before finally placing them in the normal upright orientation for 20 additional minutes. Subsequently, any unattached cells were washed with fresh media. Inlet and outlet ports were filled with cell media and the petri dish was placed back into the incubator. The fibronectin coating coupled with the rotation steps ensured HUVECs attachment to the lumen surface. This was confirmed through phase contrast images of the top, midplane, and bottom planes of the microvessel as shown in Supplementary Figure 2.

    We have also included the phase contrast images of the bottom, midplane and top parts of the microvessel for Day 0, Day 1, Day 3 and Day 5 in Supplementary Figure 2. These timelapse images are visual feedback in ensuring proper cell attachment during the course of experiments.
  1. The inlet setup is well described, but what about the outlet? How is pressure controlled at the outlet? Is there a method to prevent backflow and keep the system running smoothly? Also, what steps were taken to prevent contamination over long culture periods?
    We thank the reviewer for acknowledging that the inlet setup is described well. We also appreciate the reviewer’s insightful questions. We refer the reviewer to the revised text in section 2.5 (page 7, line 263), highlighted in yellow:

    A slit was cut at the outlet port of the microdevice using a sterile cutting blade to achieve zero-gauge pressure at the outlet port. This is analogous to grounding in electrical circuits. 100 μL of cell media was added to the slit to form a continuous fluid circuit. This enabled pooling of perfused media from the slit onto the glass slide and surrounding petri dish, thereby ensuring continuous flow and negligible backpressure. The PPA and the microdevice were housed in a deep petri dish (Fisher Scientific, FB0875711), and wrapped with parafilm (Fisher Scientific, S37440) before placing in the 37°C incubator, thereby minimizing contamination.